# Nonlinear Rheological Processes Modeling in Three-Layer Plates with a Polyurethane Foam Core

**DOI:** 10.3390/polym14102093

**Published:** 2022-05-20

**Authors:** Anton Chepurnenko

**Affiliations:** Strength of Materials Department, Faculty of Civil and Industrial Engineering, Don State Technical University, Rostov-on-Don 344000, Russia; anton_chepurnenk@mail.ru; Tel.: +7-8632-019136

**Keywords:** polyurethane foam, sandwich panels, three-layer plate, creep, Maxwell–Gurevich equation, numerical simulation, nonlinearity

## Abstract

Introduction: Three-layer structures with a polyurethane foam filler are widely used in construction as roofing and wall panels. The purpose of this work is to develop a method for calculating the bending of three-layer plates with a polyurethane foam filler, taking into account the nonlinear creep of the middle layer. The non-linear Maxwell–Gurevich equation is used as the polyurethane foam creep law. Methods: In the article, the system of resolving the equations is obtained, and the solution is carried out numerically by the finite difference method in combination with the Euler method in a MATLAB environment. An analytical solution is also obtained for a plate hinged along the contour. Results: The developed model and calculation algorithms are verified by comparison with the calculation in the ANSYS software package. A comparison with the calculation according to the linear theory is also carried out, and the effects caused by the non-linear creep of polyurethane foam are revealed. Conclusion: It has been established that when nonlinear creep is taken into account, in contrast to the linear law, the stresses in the plate are not constant in time. In the faces, at the initial stage, the stresses increase with a subsequent return to the initial values, and in the filler, on the contrary, the stresses at the initial stage decrease. These results indicate the need to take into account the nonlinear creep of polyurethane foam in the calculation of sandwich panels.

## 1. Introduction

Three-layer plates in which the outer layers are made of materials with high physical and mechanical characteristics (steel, aluminum, and fiberglass) and the middle layer consists of light aggregate are widely used in construction in the form of roofing and wall panels [1]. Such structures are much lighter than single-layer ones with the same strength and rigidity. Mineral wool and foamed polymers are usually used as the core of three-layer plates. Polyurethane foam is one of the most common polymer fillers for sandwich panels. A significant advantage of this material is the one-stage fabrication of structures by spraying or pouring.

At the same time, polyurethane foam, like all polymers, has rheological properties that lead to an increase in sandwich panel deformations over time [2]. Due to the low deformation characteristics of polyurethane foam compared to the materials used in the outer layers of sandwich panels, the classical theory of plates is not applicable to such structures [3].

One of the first publications on the creep of three-layer panels with a polyurethane foam core belongs to J.S. Huang and L.G. Gibson [4]. In this paper, sandwich panels with aluminum faces were considered. To describe the creep of polyurethane foam, the linear theory of viscoelasticity was used. In the work of C. Chen et al. [5], the steady creep of sandwich panels with a middle layer of aluminum foam was considered at various levels of stress and temperature using the Timoshenko beam model [6]. Subsequently, Timoshenko’s theory of beams was used in the works of M. Garrido et al. to predict the creep of three-layer beams with a middle layer of polyurethane foam [7] and PET foam [8].

Y. Frostig et al. [9] proposed a refined high-order theory for the calculation of multilayer beams. The development of this theory for calculations, taking into account creep, was carried out in the article by M. Ramezani and E. Hamed [10]. In the work of E. Hamed and Y. Frostig [11], in addition to creep, geometric nonlinearity was also taken into account. Various formulations of the three-layer beams theories and finite elements for their calculation are also presented in [12,13,14,15,16,17].

Most of the work on the calculation of three-layer structures with a polyurethane foam filler, taking into account creep, refer to three-layer beams. However, three-layer plates that are subjected to bending in two planes are of no less interest. In addition, many experimental studies, including [18,19,20], describe the creep of polyurethane foam using a power law in which time is present in an explicit form. The disadvantage of this law is the impossibility of its application for complex loading conditions when the load is a function of time.

For many polymers, creep curves are well described by the generalized non-linear Maxwell–Gurevich equation [21], which has the form:(1)$$\frac{\partial \varepsilon_{ij}^{cr}}{\partial t}=\frac{f_{ij}^{*}}{\eta^{*}},\;\;i=x,y,z,\;\mathrm{and}\;\;j=x,y,z,$$
where $f_{ij}^{*}$ is the stress function and $\eta^{*}$ is the relaxation viscosity; and
(2)$$f_{ij}^{*}=\frac{3}{2}\left(\sigma_{ij}-p\delta_{ij}\right)-E_{\infty }\varepsilon_{ij}^{cr},\;\eta^{*}=\eta_{0}^{*}\;\exp\left(-\frac{\left|f_{max}^{*}\;\right|}{m^{*}}\;\right),\;\mathrm{and}\;f_{max}^{*}={\left|\frac{3}{2}\left(\sigma_{rr}-p\;\right)-E_{\infty }\varepsilon_{rr}^{cr}\right|}_{max},$$
where $\delta_{ij}$ is the Kronecker symbol, $p=\left(\sigma_{x}+\sigma_{y}+\sigma_{z}\right)/3$ is the mean stress, $m^{*}$ is the velocity module, $\eta_{0}^{*}$ is the initial relaxation viscosity, and $E_{\infty }$ is the high elasticity modulus, and the rr indexes correspond to the directions of principal stresses.

The possibility of using this equation to describe the creep of polyurethane foam was shown in [22].

The purpose of this work is to obtain resolving equations for the calculation of three-layer plates with a polyurethane foam core for arbitrary creep laws, including nonlinear ones, as well as to develop algorithms for their solution.

## 2. Materials and Methods

### 2.1. Derivation of the Resolving Equations

A three-layer plate with a total thickness *h* (mm) and thickness of the outer layers δ (mm) is considered. We assume that δ is small compared to *h*, and the distance between the median planes of the outer layers can be approximately taken as equal to *h*. Since the modulus of elasticity of light filler, as a rule, is significantly lower than the modulus of elasticity of the faces, it can be assumed that bending and torsional moments are completely perceived by the faces, and the middle layer works only on transverse shear, perceiving the transverse force. An infinitesimal plate element with stresses acting in it is shown in Figure 1. The indices “l” correspond to the lower face, “up” to the upper face, and “m” to the middle layer.

The displacements of the points of the upper face will be denoted by $u^{up},\;v^{up},\;\mathrm{and}\;w^{up}$, and the displacements of the lower face will be denoted by $u^{l},\;v^{l},\;\mathrm{and}\;w^{l}$. The middle layer will be assumed to be incompressible in the vertical direction ($w^{l}=w^{up}=w$). The deformations of the upper and lower faces are calculated as follows:(3)$$\varepsilon_{x}^{up\left(l\right)}=\frac{\partial u^{up\left(l\right)}}{\partial x};\;\;\varepsilon_{y}^{up\left(l\right)}=\frac{\partial v^{up\left(l\right)}}{\partial y};\;\gamma_{xy}^{up\left(l\right)}=\frac{\partial u^{up\left(l\right)}}{\partial y}+\frac{\partial v^{up\left(l\right)}}{\partial x}.$$

Elastic work is accepted for the faces. The deformations in them are related to stresses as follows:(4)$$\varepsilon_{x}^{up\left(l\right)}=\frac{1}{E}\left(\sigma_{x}^{up\left(l\right)}-\nu \sigma_{y}^{up\left(l\right)}\right);\;\varepsilon_{y}^{up\left(l\right)}=\frac{1}{E}\left(\sigma_{y}^{up\left(l\right)}-\nu \sigma_{x}^{up\left(l\right)}\right);\;\mathrm{and}\;\gamma_{xy}^{up\left(l\right)}=\frac{2\left(1+\nu \right)\tau_{xy}^{up\left(l\right)}}{E};$$
or, in inverse form:(5)$$\begin{array}{l} \sigma_{x}^{up\left(l\right)}=\frac{E}{1-\nu^{2}}\left(\varepsilon_{x}^{up\left(l\right)}+\nu \varepsilon_{y}^{up\left(l\right)}\right);\;\\ \sigma_{y}^{up\left(l\right)}=\frac{E}{1-\nu^{2}}\left(\varepsilon_{y}^{up\left(l\right)}+\nu \varepsilon_{x}^{up\left(l\right)}\right);\;\;\\ \tau_{xy}^{up\left(l\right)}=\;\frac{E}{2\left(1+\nu \right)}\gamma_{xy}^{up\left(l\right)}. \end{array}\mathrm{and}$$

For the displacements of the middle layer, we will assume a linear distribution over the thickness:(6)$$u^{m}=\frac{u^{up}+u^{l}}{2}+\frac{u^{l}-u^{up}}{h}z=u+\alpha z,\;\mathrm{and}\;v^{m}=\frac{v^{up}+v^{l}}{2}+\frac{v^{l}-v^{up}}{h}z=v+\beta z,$$
where u and v are the midplane displacements.

The total shear strains of the middle layer are determined from the Cauchy relations:(7)$$\begin{array}{l} \gamma_{zx}^{m}=\frac{\partial u^{m}}{\partial z}+\frac{\partial w}{\partial x}=\alpha +\frac{\partial w}{\partial x},\;\mathrm{and}\\ \gamma_{zy}^{m}=\frac{\partial v^{m}}{\partial z}+\frac{\partial w}{\partial y}=\beta +\frac{\partial w}{\partial y}. \end{array}$$

On the other hand, they represent the sum of elastic and creep deformations:(8)$$\gamma_{zx}^{m}=\frac{\tau_{zx}^{m}}{G_{m}}+\gamma_{zx}^{cr};\;\gamma_{zy}^{m}=\frac{\tau_{zy}^{m}}{G_{m}}+\gamma_{zy}^{cr},$$
where $G_{m}$ is the middle layer shear modulus (MPa).

Taking into account (7) and (8), the shear stresses in the middle layer can be written as:(9)$$\begin{array}{l} \tau_{zx}^{m}=G_{m}\left(\gamma_{zx}^{m}-\gamma_{zx}^{cr}\right)=G_{m}\left(\alpha +\frac{\partial w}{\partial x}-\gamma_{zx}^{cr}\right),\;\mathrm{and}\;\\ \tau_{zy}^{m}=G_{m}\left(\gamma_{zy}^{m}-\gamma_{zy}^{cr}\right)=G_{m}\left(\beta +\frac{\partial w}{\partial y}-\gamma_{zy}^{cr}\right).\;\; \end{array}$$

For the shear stresses along the thickness of the middle layer, a uniform distribution is assumed. In this case, the shear forces will be written in the form:(10)$$\begin{array}{l} Q_{x}=\tau_{zx}^{m}h=G_{m}h\left(\alpha +\frac{\partial w}{\partial x}-\gamma_{zx}^{cr}\right),\;\mathrm{and}\;\;\\ Q_{y}=\tau_{zy}^{m}h=G_{m}h\left(\beta +\frac{\partial w}{\partial y}-\gamma_{zy}^{cr}\right). \end{array}$$

Bending and torsional moments are related to the stresses in the faces in the following way:(11)$$M_{x}=\left(\sigma_{x}^{l}-\sigma_{x}^{up}\right)\cdot \delta \cdot \frac{h}{2},\;M_{y}=\left(\sigma_{y}^{l}-\sigma_{y}^{up}\right)\cdot \delta \cdot \frac{h}{2},\;\mathrm{and}\;M_{xy}=\left(\tau_{xy}^{l}-\tau_{xy}^{up}\right)\cdot \delta \cdot \frac{h}{2}.$$

Substituting (3) into (5), and then (5) into (11), we get:(12)$$M_{x}=D\left(\frac{\partial \alpha }{\partial x}+\nu \frac{\partial \beta }{\partial y}\right),\;M_{y}=D\left(\nu \frac{\partial \alpha }{\partial x}+\frac{\partial \beta }{\partial y}\right),\;\mathrm{and}\;M_{xy}=\frac{D\left(1-\nu \right)}{2}\left(\frac{\partial \alpha }{\partial y}+\frac{\partial \beta }{\partial x}\right),$$
where $D=\frac{E\delta h^{2}}{2\left(1-\nu^{2}\right)}$ is the cylindrical stiffness of a three-layer plate (kN·m).

Internal forces in the plate are related by the differential equilibrium equations:(13)$$\frac{\partial Q_{x}}{\partial x}+\frac{\partial Q_{y}}{\partial y}=-q,\;\frac{\partial M_{x}}{\partial x}+\frac{\partial M_{xy}}{\partial y}=Q_{x},\;\mathrm{and}\;\frac{\partial M_{xy}}{\partial x}+\frac{\partial M_{y}}{\partial y}=Q_{y}.$$

Equation (13) can be reduced to one equation for the bending and torque moments:(14)$$\frac{\partial^{2}M_{x}}{\partial x^{2}}+2\frac{\partial^{2}M_{xy}}{\partial x\partial y}+\frac{\partial^{2}M_{y}}{\partial y^{2}}=-q.\;$$

Substituting (12) into (14), we get:(15)$$D\nabla^{2}F=-q,$$
where $F=\frac{\partial \alpha }{\partial x}+\frac{\partial \beta }{\partial y}$ is the displacement function.

Equation (15) shows that the displacement function under a constant load does not depend on time.

Next, we substitute (10) into the first Equation (13). As a result, we obtain the following differential equation:(16)$$\nabla^{2}w=-\frac{q}{G_{m}h}-F+\frac{\partial \gamma_{zx}^{cr}}{\partial x}+\frac{\partial \gamma_{zy}^{cr}}{\partial y}.$$

To calculate internal forces and stresses, it is also necessary to determine the functions *α* and *β*. To obtain resolving equations for *α* and *β*, we substitute (12) into the second and third Equation (13):(17)$$Q_{x}=\frac{\partial M_{x}}{\partial x}+\frac{\partial M_{xy}}{\partial y}=D\left(\frac{\partial^{2}\alpha }{\partial x^{2}}+\frac{1+\nu }{2}\frac{\partial^{2}\beta }{\partial x\partial y}+\frac{1-\nu }{2}\frac{\partial^{2}\alpha }{\partial y^{2}}\right),\;\mathrm{and}\;Q_{y}=\frac{\partial M_{y}}{\partial y}+\frac{\partial M_{xy}}{\partial x}=D\left(\frac{\partial^{2}\beta }{\partial y^{2}}+\frac{1+\nu }{2}\frac{\partial^{2}\alpha }{\partial x\partial y}+\frac{1-\nu }{2}\frac{\partial^{2}\beta }{\partial x^{2}}\right).$$

Using the displacement function, we eliminate the function *β* from the first equality in (17), and the function *α* from the second:(18)$$Q_{x}=\frac{D}{2}\left(\left(1-\nu \right)\nabla^{2}\alpha +\left(1+\nu \right)\frac{\partial F}{\partial x}\;\right),\;\mathrm{and}\;Q_{y}=\frac{D}{2}\left(\left(1-\nu \right)\nabla^{2}\beta +\left(1+\nu \right)\frac{\partial F}{\partial y}\right).$$

Equating the right sides of (18) and (10), we obtain the resolving equations for *α* and *β*, respectively, as follows:(19)$$\nabla^{2}\alpha -\frac{2G_{m}h}{D\left(1-\nu \right)}\alpha =\frac{2G_{m}h}{D\left(1-\nu \right)}\left(\frac{\partial w}{\partial x}-\gamma_{zx}^{cr}\right)-\frac{1+\nu }{1-\nu }\frac{\partial F}{\partial x},\;\mathrm{and}\;\nabla^{2}\beta -\frac{2G_{m}h}{D\left(1-\nu \right)}\beta =\frac{2G_{m}h}{D\left(1-\nu \right)}\left(\frac{\partial w}{\partial y}-\gamma_{zy}^{cr}\right)-\frac{1+\nu }{1-\nu }\frac{\partial F}{\partial y}.$$

Thus, the differential Equations (15), (16), and (19) completely determine the stress–strain state of a three-layer plate during bending. One can solve them sequentially.

Let us consider the boundary conditions for a plate hinged along the contour (Figure 2). On each boundary, for the functions *w*, *F*, *α*, and *β*, one boundary condition is specified. With hinged support, the deflection and bending moments on the contour are equal to zero. On the edges x=0 and x=a, we set *β* = 0, then the derivative $\frac{\partial \beta }{\partial y}$ automatically vanishes on these edges. Then, to ensure that the bending moments are equal to zero, it is necessary that the derivative $\frac{\partial \alpha }{\partial x}$ also vanishes. Similarly, on the edges y=0 and y=b, we set α=0. Then, on these edges, $\frac{\partial \alpha }{\partial x}=0$ and $\frac{\partial \beta }{\partial y}=0$. Finally, the boundary conditions can be written in the form:(20)$$at\;x=0,\;x=a:w=0,\;F=0,\;\frac{\partial \alpha }{\partial x}=0,\;\beta =0,\;\mathrm{and}\;at\;y=0,\;y=b:w=0,\;F=0,\;\alpha =0,\frac{\partial \beta }{\partial y}=0.$$

### 2.2. Calculation Algorithm

Equations (15), (16), and (19) can be solved numerically by the finite difference method. A grid is introduced in time *t* and coordinates *x* and *y*. With a load constant in time, the displacement function *F* does not depend on time, and the solution of Equation (15) is performed once. The remaining equations are solved at each time step.

At the first step, when t=0, $\gamma_{zx}^{cr}=0$ and $\gamma_{zy}^{cr}=0$. Further, after solving Equations (16) and (19), we can find the stresses in the middle layer. If the creep law is given in differential form, the creep strains at time t plus Δt can be found from the strains and stresses at time *t* using the Euler method. It is possible to use schemes of a higher order of accuracy, for example, the fourth order Runge–Kutta method.

The described algorithm was implemented by us in a MATLAB environment.

### 2.3. Analytical Solution for the Moment of the Beginning and End of the Creep Process

Equations (15), (16), and (19) with the boundary conditions (20) at *t* = 0 can also be solved analytically using double trigonometric series. The functions *F*, *w*, α, and *β* will be sought in the form:(21)$$F\left(x,y\right)=\overset{\infty }{\underset{m=1}{\sum }}\overset{\infty }{\underset{n=1}{\sum }}F_{mn}\sin\frac{m\pi x}{a}\sin\frac{n\pi y}{b};\;w\left(x,y\right)=\overset{\infty }{\underset{m=1}{\sum }}\overset{\infty }{\underset{n=1}{\sum }}w_{mn}\sin\frac{m\pi x}{a}\sin\frac{n\pi y}{b};\;\alpha \left(x,y\right)=\sum_{m=1}^{\infty }\sum_{n=1}^{\infty }\alpha_{mn}\cos\frac{m\pi x}{a}\sin\frac{n\pi y}{b};\;\mathrm{and}\;\beta \left(x,y\right)=\sum_{m=1}^{\infty }\sum_{n=1}^{\infty }\beta_{mn}\sin\frac{m\pi x}{a}\cos\frac{n\pi y}{b}.$$

The given functions satisfy the boundary conditions (20). The load function q(x,y) is expanded into a double series:(22)$$q\left(x,y\right)=\overset{\infty }{\underset{m=1}{\sum }}\overset{\infty }{\underset{n=1}{\sum }}q_{mn}\sin\frac{m\pi x}{a}\sin\frac{n\pi y}{b}.$$

The expansion coefficients are determined by the formula:(23)$$q_{mn}=\frac{4}{ab}\overset{a}{\underset{0}{\int }}\overset{b}{\underset{0}{\int }}q\left(x,y\right)\sin\frac{m\pi x}{a}\sin\frac{n\pi y}{b}dxdy.$$

In the case of a load uniformly distributed over the area (q=const):(24)$$q_{mn}=\{\begin{array}{c} \frac{16q}{mn\pi^{2}},\;\mathrm{if}\;m\;\mathrm{and}\;n\;\mathrm{are}\;\mathrm{odd}\\ 0,\;\mathrm{if}\;m\;\mathrm{or}\;n\;\mathrm{is}\;\mathrm{even} \end{array}$$

Substituting (21) and (22) into (15), (16), and (19), after transformations, we obtain formulas for the coefficients $F_{mn}$, $w_{mn}$, $\alpha_{mn}$, and $\beta_{mn}$:(25)$$F_{mn}=\frac{q_{mn}}{\pi^{2}D\left(\frac{m^{2}}{a^{2}}+\frac{n^{2}}{b^{2}}\right)};$$
(26)$$w_{mn}=\frac{q_{mn}}{\pi^{2}G_{m}h\left(\frac{m^{2}}{a^{2}}+\frac{n^{2}}{b^{2}}\right)}+\frac{q_{mn}}{\pi^{4}D{\left(\frac{m^{2}}{a^{2}}+\frac{n^{2}}{b^{2}}\right)}^{2}};$$
(27)$$\alpha_{mn}=-\frac{a^{3}b^{4}m\cdot q_{mn}}{\pi^{3}D{\left(a^{2}n^{2}+b^{2}m^{2}\right)}^{2}};\;\mathrm{and}\;\beta_{mn}=-\frac{a^{4}b^{3}n\cdot q_{mn}}{\pi^{3}D{\left(a^{2}n^{2}+b^{2}m^{2}\right)}^{2}}.$$

In formula (26), the first term represents the contribution of the shear deformations of the middle layer to the deflection of the plate, and the second term is the contribution of the faces’ deformations.

It is shown in [23] that if the material of the structure obeys the Maxwell–Gurevich equation, then to obtain a solution at *t*→∞, it is sufficient to replace the instantaneous elastic constants with long-term ones in the solution at *t* = 0. The long-term shear modulus of the filler $G_{long}$ can be calculated by the formula:(28)$$G_{long}=\frac{G\cdot G_{\infty }}{G+G_{\infty }},$$
where G is the instant shear modulus, $G_{\infty }=E_{\infty }/3$.

It can be seen from formula (27) that the coefficients $\alpha_{mn}$ and $\beta_{mn}$, and hence the functions *α* and *β*, do not depend on the shear modulus of the middle layer. This means that at *t*→∞, the bending and torque values determined by formula (12) will be the same as at *t* = 0. The same applies to shear forces, as can be seen from formula (17).

## 3. Results

The first step to test the developed technique was to solve a test problem for a square plate with steel faces, followed by comparison with the ANSYS software package. The calculation was carried out with the following initial data: a plate thickness of *h* = 80 mm, a modulus of elasticity of faces of *E* = 2 × 10^5^ MPa, a Poisson’s ratio of faces of *ν* = 0.3, their thickness δ = 1.5 mm, slab dimensions of a=b=3000 mm, a middle layer shear modulus of $G_{m}=12.6\;\mathrm{MPa}$, and a uniformly distributed load on the plate of q=6 kPa. The rheological parameters of the polyurethane foam are: $E_{\infty }=27.38\;\mathrm{MPa}$, $m^{*}$ = 0.0218 MPa, and $\eta_{0}^{*}=5.15\times 10^{7}$ MPa·s.

In ANSYS, the middle layer was modeled with SOLID186 3D finite elements, and the faces were modeled with Surface Coating (SURF154) elements. To specify a custom creep law, the User Programmable Features extension was used. The Maxwell–Gurevich law was implemented as the usercreep.f subroutine in the FORTRAN language.

Figure 3 shows the graphs of the deflection growth in the center of the plate, obtained by the author’s method and in the ANSYS program.

Table 1 shows the values of the deflection in the center of the plate at time *t* = 0 and *t* = 800 h when solving Equations (15), (16), and (19) by the finite difference method for a different number of intervals, $n=n_{x}=n_{y}$, in *x* and *y.*

The analytical value of the deflection at t=0, obtained using four terms of the series (m=1, 3 and n=1, 3), was 11.9 mm, which differs from the solution in ANSYS by 2.6%. The maximum discrepancy between the results at *t* = 800 h is 3.9%. The displacement isofields obtained in ANSYS at *t* = 800 h are shown in Figure 4.

To analyze the effect of nonlinear creep on the stress–strain state of three-layer plates, we performed a calculation for the above initial data with different values of load *q* from 2 to 10 kPa. The resulting deflection growth curves are shown in Figure 5. For comparison, a calculation was performed using a linearized equation ($\eta^{*}=\eta_{0}^{*}=const$) which coincides in structure with the linear Maxwell–Thompson equation. The corresponding plots are shown with dashed lines.

Figure 5 shows that with increasing load, the discrepancy between the results obtained using the nonlinear and linearized creep equations increases. According to the nonlinear theory, the increase in deformations and the attenuation of the rate of the deflection growth occur faster.

Another effect due to non-linear creep is the time variability of stresses in the faces and core under high loads. Figure 6 and Figure 7 show graphs of the change in time of the maximum tangential stresses in the lower skin and the middle layer at *q* = 10 kPa. In the middle layer, the stresses decrease at the initial stage, and they increase in the faces. At *t*→∞, as shown earlier in Section 2, there is a return to the elastic solution. According to the linear theory, in contrast to the nonlinear one, the stresses in all layers under a constant load do not change over time.

## 4. Discussion

The established differences in the nature of the change in deflections with time when using the linear and nonlinear creep equations are consistent with the results obtained in [22] for three-layer beams with a polyurethane foam filler. The time variability of stresses in the faces and core can be explained by the presence of unsteady creep at the initial stage, when the creep strains of the polymer lag behind the stresses in phase. Similar effects were previously revealed using the nonlinear Maxwell–Gurevich equation for other polymers in [24,25,26,27,28]. In the considered example, the increase in shear stresses in the faces turned out to be not so significant and amounted to only 7%. Under other conditions, the growth may be more noticeable and pose a danger in terms of the bearing capacity loss of the plate. Therefore, when calculating three-layer plates with a polyurethane foam filler, it is preferable to use a nonlinear equation of the relationship between stresses and creep strains.

It should be noted that for three-layer shells, in comparison with plates, a different character of the change in the stress–strain state is observed. For the plates considered in this article, the increase in displacements during creep was about 45%. For the shells, even with a slight curvature, the creep of the core does not have a noticeable effect on the deflection [29]. This can be explained by the fact that three-layer shells, unlike plates, mainly work in tension and compression, and the contribution of shear forces to their stress–strain state is small.

In the present work, a one-term version of the Maxwell–Gurevich equation is used. We note that most polymers are characterized by a discrete spectrum of relaxation times, and it is necessary to take into account at least two terms of the spectrum [30]. The resolving equations we obtained make it possible to take this circumstance into account, but for this, it is necessary to know the rheological parameters for each spectrum of the polymer. Further, the equations presented in this article make it possible to use not only the Maxwell–Gurevich equation, but also any other creep law.

This article does not touch upon the strength of the adhesive bond between skins and filler. To solve this problem, the contact layer method [31] can be applied. Some solutions for three-layer beams and plates can be found in [32,33]. However, these publications do not take into account the deformations of the transverse shear of the layers, which, for three-layer plates with a light filler, can make a more significant contribution to the deflection than to the deformations of the faces. Our further research will be aimed at modeling the adhesion between the polyurethane foam core and faces in three-layer panels, taking into account the nonlinear rheological properties of the core.

## 5. Conclusions

A system of resolving differential equations has been obtained for calculating three-layer plates with polyurethane foam filler, taking into account the nonlinear creep of the middle layer. A numerical algorithm for solving this system is proposed. An analytical solution is also obtained for a plate hinged along the contour at t=0 and t→∞. It is shown that when using the Maxwell–Gurevich equation, the stresses in all layers at t=0 and t→∞ coincide. The developed algorithm was tested by comparison with the calculation using the finite element method in the ANSYS software package. The discrepancy does not exceed 5%. Significant differences between the results when using linear and non-linear theory are revealed. When nonlinear creep is taken into account, the stresses in the skins and core are not constant in time. In the skins, at the initial stage, an increase in stresses is observed, followed by a return to the original values.

## Figures and Tables

**Figure 1 polymers-14-02093-f001:**
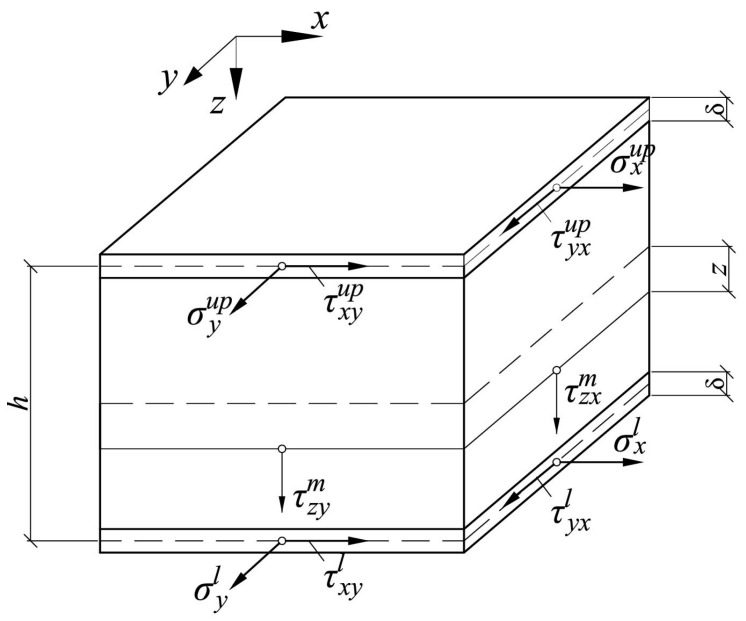
Infinitesimal plate element.

**Figure 2 polymers-14-02093-f002:**
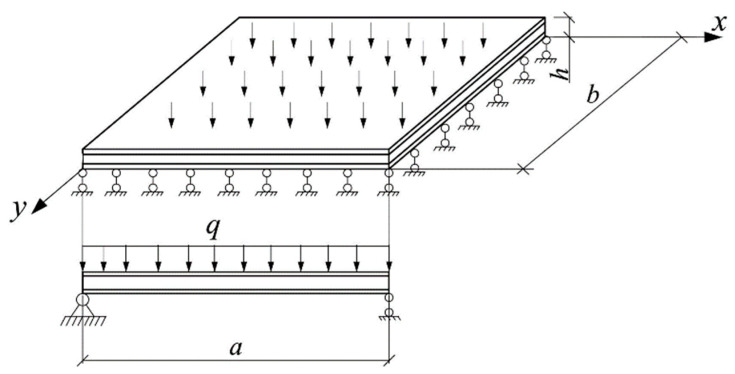
Calculation scheme for the plate hinged along the contour.

**Figure 3 polymers-14-02093-f003:**
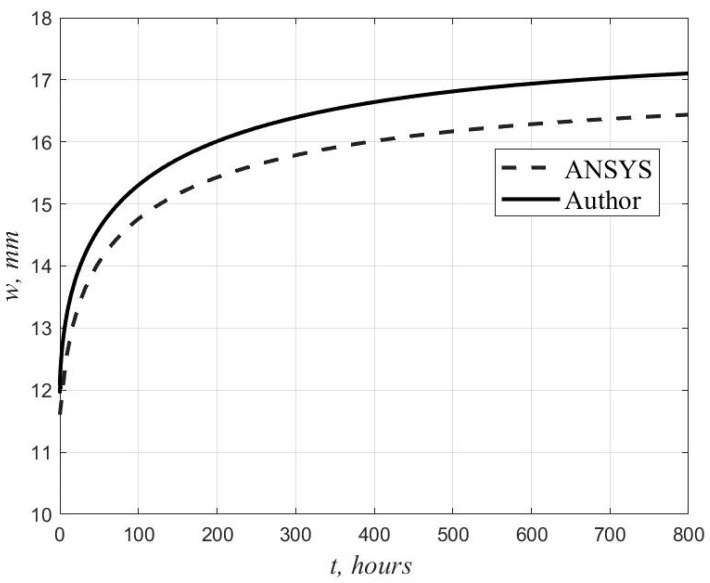
Comparison of the author’s solution with the results in the ANSYS software package.

**Figure 4 polymers-14-02093-f004:**
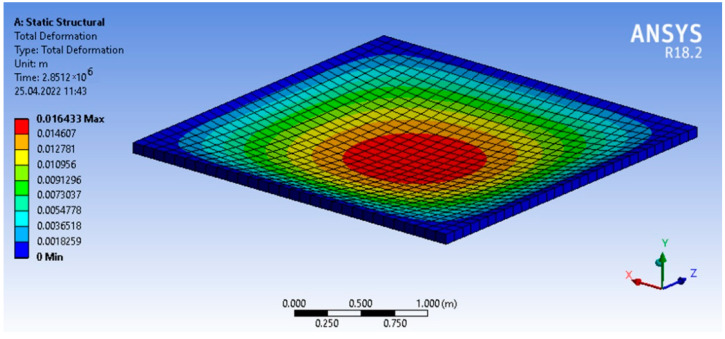
Displacement isofields in ANSYS at *t* = 800 h.

**Figure 5 polymers-14-02093-f005:**
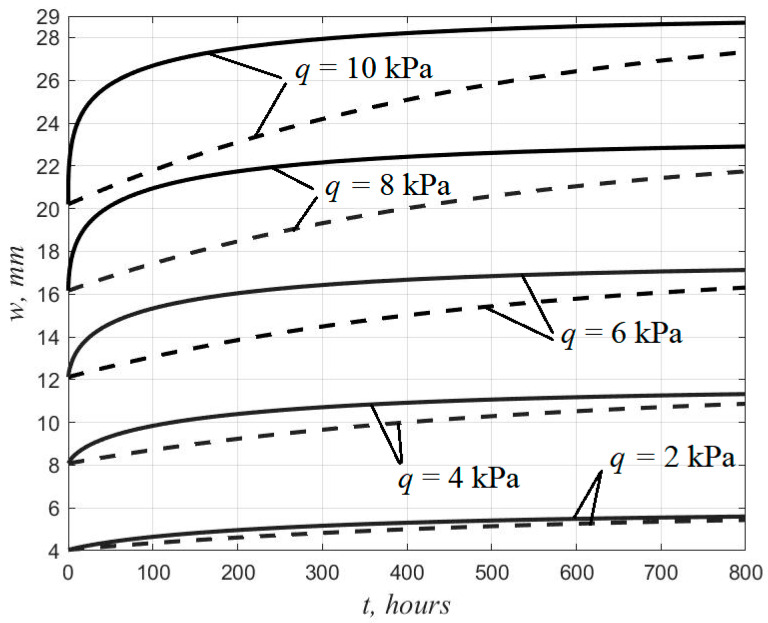
Graphs of the deflection growth according to the nonlinear and linearized theory at various load values.

**Figure 6 polymers-14-02093-f006:**
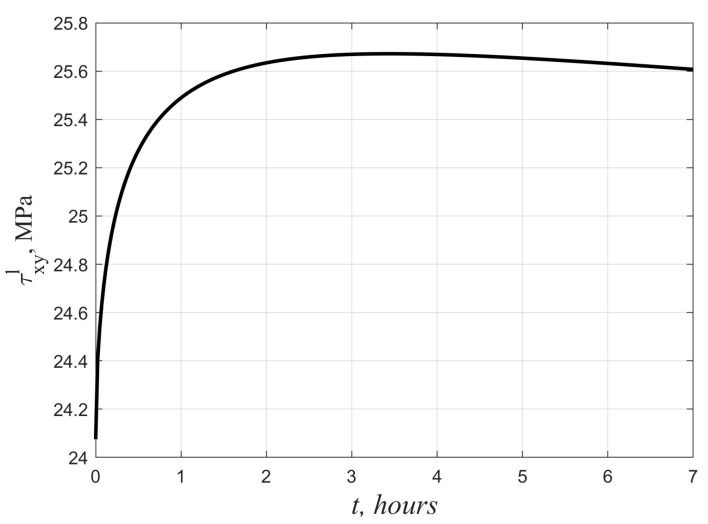
Change in time of maximum shear stresses in the faces.

**Figure 7 polymers-14-02093-f007:**
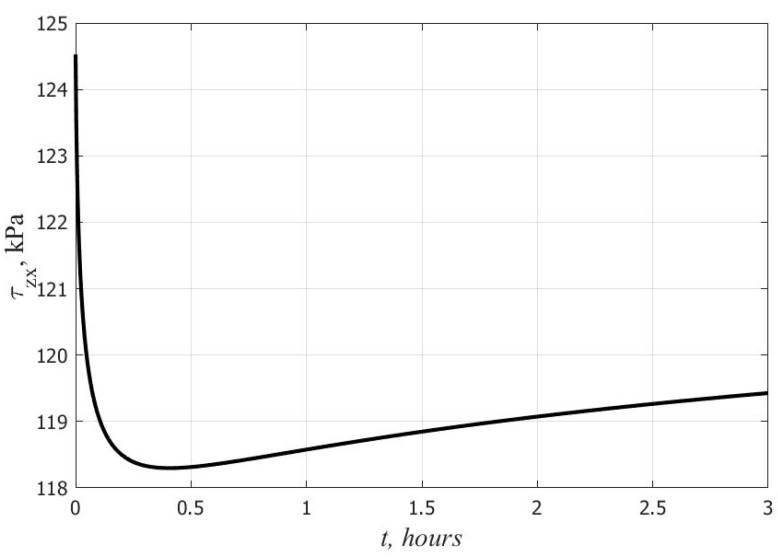
Change in time of maximum shear stresses in the core.

**Table 1 polymers-14-02093-t001:** Mesh sensitivity analysis when using the finite difference method.

*n*	4	6	10	20	40	80
*w*_0_, mm	11.64	11.90	12.04	12.10	12.12	12.12
*w*_800_, mm	16.90	17.12	17.15	17.12	17.10	17.10

## Data Availability

Not applicable.

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
