# Peer review of "Nonlinear Rheological Processes Modeling in Three-Layer Plates with a Polyurethane Foam Core"

_polymers, 2022, doi:10.3390/polym14102093_

Round 1

Reviewer 1 Report

Dear author,

units are missing e.g. h thickness (row 74), Gm shear module (row 97), cylindrical stiffness (row 104) etc. 

Chapter results: row 173: h= 8 cm  use the unit mm, not cm.  Row 175 ?=?=3 m not m but mm. 

Figure 4 - legend - you are as one author - use ANSYS and Author (not Authors).

Change description: Figure 4 to Figure 3 (the order of the figures is one larger from figure 4 onwards, because figure 3 is missing), Figure 5 to Figure 4 etc.  and in text too: e.g. row 182 Figure 4 shows graphs ... change to Figure 3 shows graphs ... etc.    Row 187 .... are shown in Fig. 5. change to are shown in Fig. 4. etc. row 195 ...........

Row 185: ANSYS by 2.6% change to NSYS by 2.6SPACE%. etc. row 186:  ... is 3.9%. change to 3.9SPACE%.  and e.g. rows 227, 246.   Publications 3 (from 1988) and 6 (from 1965) and 19 - do not use publications older than 20 years. You can used only if there's something in the publication that hasn't been published anywhere else.   I recommend the publication accepted after minor revision. Have a nice day.

Reviewer 2 Report

The manuscript polymers-1726799 was reviewed. The manuscript is prepared good. However, the following comments must be considered before   making the final decision:

  1. The paper contains many equations. The authors are suggested to reduce them and instead focus on the scientific discussion of them.
  2. The Discussion section is prepared short. The authors must provide some of comparative analysis with the previous research studies in this section.
  3. The authors are suggested to add the name of boundary condition in Fig.2.
  4. Was any mesh sensitivity analysis performed before the numerical simulation? If so, the authors are suggested to provide the related information in the manuscript.
